# Tumors of the Parapharyngeal Space Presenting with Obstructive Sleep Apnea: A Case Report and Literature Review

**DOI:** 10.3390/jpm15080331

**Published:** 2025-07-28

**Authors:** Luca Cerri, Francesco Giombi, Michele Cerasuolo, Gian Marco Pace, Anna Losurdo, Giuseppe Lunardi, Francesco Grecchi, Elena Volpini, Luca Malvezzi

**Affiliations:** 1Otorhinolaryngology-Head & Neck Surgery Unit, IRCCS Humanitas Research Hospital, Via Manzoni 56, Rozzano, 20089 Milan, Italy; luca.cerri@humanitas.it (L.C.); gianmarco.pace@humanitas.it (G.M.P.); luca.malvezzi@humanitas.it (L.M.); 2Department of Biomedical Sciences, Humanitas University, Via Rita Levi Montalcini 4, Pieve Emanuele, 20090 Milan, Italy; 3Otorhinolaryngology-Head & Neck Surgery Unit, Casa di Cura Humanitas San Pio X, Via Francesco Nava 31, 20159 Milan, Italy; michele.cerasuolo@humanitas.it (M.C.); giuseppe.lunardi@sanpiox.humanitas.it (G.L.); francesco.grecchi@humanitas.it (F.G.); 4Neurology and Neurophysiopathology Unit, Casa di Cura Humanitas San Pio X, Via Francesco Nava 31, 20159 Milan, Italy; anna.losurdo@sanpiox.humanitas.it; 5General Medicine Unit, Casa di Cura Humanitas San Pio X, Via Francesco Nava 31, 20159 Milan, Italy; elena.volpini@sanpiox.humanitas.it

**Keywords:** obstructive sleep apnea, sleep-disordered breathing, parapharyngeal space, upper aerodigestive tract tumors, pleomorphic adenoma

## Abstract

**Introduction**: Obstructive sleep apnea syndrome (OSAS) is caused by anatomical and non-anatomical factors which lead to upper airway (UA) obstruction during sleep. Intrinsic UA collapse is the most frequent determinant of OSA. In the era of personalized medicine, adopting a tailored diagnostic approach is essential to rule out secondary causes of UA collapse, particularly those stemming from extrinsic anatomical factors. Although being rarely considered in the differential diagnosis, space-occupying lesions of deep cervical spaces such as the parapharyngeal space (PPS) may be responsible for airway obstruction and lead to OSAS. **Objective**: This study aimed to present an atypical case of OSAS caused by extrinsic PPS compression, outlining the relevance of modern personalized medicine in the diagnostic and therapeutic protocols, and to enhance understanding through a comprehensive literature review. **Methods**: A 60-year-old female presented with sleep-disordered complaints and was diagnosed with severe OSAS after polysomnography. At physical examination, a swelling of the right posterior oropharyngeal mucosa was noticed. Imaging confirmed the suspicion of a PPS tumor, and transcervical resection was planned. Case presentation was adherent to the CARE checklist. A comprehensive literature review was conducted using the most reliable scientific databases. **Results**: Surgery was uneventful, and the patient made a full recovery. The histopathology report was consistent with the diagnosis of pleomorphic adenoma. Postoperative outcomes showed marked improvement in polysomnographic parameters and symptom burden. **Conclusions**: Parapharyngeal space tumors are a rare, often overlooked cause of OSA. This case highlights the role of a personalized head and neck assessment in OSA patients, particularly in identifying structural causes and offering definitive surgical management when indicated.

## 1. Introduction

Obstructive sleep apnea (OSA) is a complex and highly prevalent disorder characterized by recurrent upper airway (UA) collapse during sleep, leading to frequent nocturnal arousals, the fragmentation of sleep, and intermittent oxygen desaturation [1]. Despite often being underrecognized, OSA has been shown to affect a substantial proportion of the general population, with some studies reporting that up to 25% of women and 50% of men experience moderate-to-severe forms of the disease [2,3,4]. Importantly, individuals with OSA are consistently found to be at elevated risk for a wide range of systemic and potentially life-threatening conditions, including cardiovascular diseases such as arterial hypertension, coronary artery disease, and cerebrovascular accidents, and metabolic disorders such as diabetes mellitus and metabolic syndrome, as well as neurodegenerative diseases, thereby underscoring its significance as a major public health concern [5,6,7].

The pathophysiology of OSA involves complex interactions between structural, neuromuscular, and neurophysiological factors [8]. Common risk factors include obesity, craniofacial abnormalities, adenotonsillar hypertrophy, and increased neck circumference [9]. While OSA is most often due to anatomical or functional airway alterations, space-occupying lesions, such as parapharyngeal space tumors (PSTs), can also cause airway obstruction and present with sleep-disordered breathing [10]. As we enter the era of personalized medicine, a tailored approach to a widespread condition like OSA is essential to effectively distinguish between anatomical and non-anatomical determinants of the disease. Furthermore, in this context, a personalized physical examination, as well as a detailed assessment of polysomnographic parameters, is essential to rule out possible extrinsic factors responsible for UA collapse during sleep.

Parapharyngeal space tumors are rare conditions, accounting for less than 1% of head and neck neoplasms. They are most often benign tumors originating from the salivary glands, peripheral nerves, or paraganglionic tissue [11]. As these tumors enlarge, they can compress the oropharyngeal airway, promoting collapse during sleep. Reported symptoms may include snoring, witnessed apneas, and daytime sleepiness, which can delay diagnosis [12].

Despite occasional reports, the relationship between PSTs and OSA remains poorly explored in the literature. We herein present a case of a PST presenting with severe OSA symptoms, with dramatic improvement following surgical resection, and review the current evidence on this rare clinical presentation.

## 2. Case Presentation

The presentation of the case was adherent to the CARE (CAse REport) guidelines [13]. A 60-year-old female patient presented with a history of worsening sleep symptoms over the preceding year. Primary complaints included loud snoring, excessive daytime sleepiness with an Epworth Sleepiness Scale (ESS) score of 12/24, reduced workplace productivity, nasal obstruction, and globus pharyngeus. Her past medical history was notable for hypertension and dyslipidemia. She reported no allergies and had never smoked. Physical examination revealed a height of 163 cm, weight of 70 kg, and body mass index (BMI) of 26.35.

A previous polysomnography (PSG) revealed severe OSA with an apnea–hypopnea index (AHI) of 77.1 events per hour (77.3 and 76.8, in supine and non-supine positions, respectively), an oxygen desaturation index (ODI) of 73.3, with a nadir oxygen saturation of 68%, and a percentage of time spent with oxygen saturation below 90% (CT90) of 24.7%. Snoring accounted for 29.3% of the total recorded time. Flexible endoscopy revealed a prominent mass in the right posterior pharyngeal wall, which caused partial obstruction of the pharyngeal lumen. On inspection of the oral cavity, a bulging, firm swelling of the right hemipalate was observed, with medialization of the palatine tonsil.

The patient was referred for magnetic resonance imaging (MRI) of the head and neck, which revealed a heterogeneous mass in the right parapharyngeal space measuring approximately 52 × 27 mm with a craniocaudal extension of 60 mm with displacement of the medial pharyngeal wall, without signs of diffusion restriction on diffusion-weighted imaging (DWI) sequences (Figure 1A–C).

The patient subsequently underwent resection of the lesion via a transcervical approach (Figure 2A). A 7 cm right-sided neck incision was performed within a natural skin crease about 3–4 cm below the mandible. Subplatysmal planes were elevated up to the level of the mandible; the sternocleidomastoid muscle was retracted laterally and the submandibular gland mobilized and retracted along with the facial vein, ensuring identification and preservation of the marginal mandibular branch of the facial nerve and the hypoglossal nerve. The tendons of the digastric muscle and stylohyoid muscle were transected to maximize exposure; the carotid sheath was identified and retracted laterally, providing adequate access to the PPS. Blunt finger dissection was then used to free the mass from its circumferential attachments, after which it was transected off the deep lobe of the parotid gland. Finally, the surgical cavity was irrigated copiously and hemostasis was achieved. A cervical suction drain was placed to prevent the risk of postoperative hematoma.

The procedure was uneventful, and the patient was discharged on postoperative day 2 with an unremarkable recovery. No postoperative neural deficits were noted on clinical examination.

A final histopathological examination confirmed the diagnosis of pleomorphic adenoma with a predominant mesenchymal chondromatous component (Figure 3).

Postoperative follow-up polysomnography was conducted at 2 months, showing significant improvement in OSA parameters, with an AHI of 43.5 (unchanged by position), an ODI of 42.5, a nadir SpO2 of 72%, a CT90 of 34.2%, and snoring accounting for 58.2% of the total sleep time. The ESS score improved to 6/24. At 4 months postoperatively, the polysomnography results showed further improvement, with an AHI of 13.7 (supine AHI 23.2, non-supine AHI 8.5), an ODI of 14.6, a nadir SpO2 of 81%, a CT90 of 17.9%, and snoring accounting for 63.4% of the total sleep time. The ESS score further decreased to 5/24, indicating a major reduction in daytime sleepiness (Table 1).

## 3. Literature Review

A comprehensive literature search was performed using the electronic databases of the Cochrane Library, MEDLINE, SCOPUS, and PubMed. Published articles from January 2000 to January 2025 were included. Included papers had to fulfill the following criteria: (i) diagnosis of a PPS neoplasm; (ii) concomitant diagnosis/symptoms of OSA; (iii) availability of a histopathologic report; (iv) age of population > 18 years; (v) English language. The following keywords were used: “Parapharynx” OR “Parapharyngeal” AND “Neoplasm” OR “Tumor” AND “Sleep apnea” OR “OSAS”. Peer-reviewed articles in press were also included. Selected papers comprised case reports or case series. To date, no observational studies nor randomized controlled trials have been published. Reviews, editorials, and letters to editors were excluded from the research, as well as studies reporting self-referred snoring with no definite symptoms of OSA. The bibliographies of all selected articles were also reviewed to find any other relevant studies. The selected studies were categorized based on their design. The following data were extracted: patient demographics, tumor origin, surgical approach, histopathological report, follow-up duration, and pre- and postoperative apnea–hypopnea index (AHI) values.

A total of 131 articles were screened for eligibility. After the exclusion of duplicates, a total of 16 studies were selected based on the inclusion criteria. Overall, 14 case reports and two case series were included, for a total of 18 patients. A detailed presentation of the included studies is shown in Table 2 [14,15,16,17,18,19,20,21,22,23,24,25,26,27,28,29]. There was a male predominance (M/F = 17/1). The mean age was 49.3 ± 17.2 years, and the mean BMI was 27.9 ± 5.4. The majority of neoplasms originated from the parapharyngeal soft tissues (*n* = 11/18, 61.1%; left-sided: 6; right-sided: 5). In seven cases (*n* = 7/18, 38.9%), tumors arose from the deep lobe of the parotid gland, extending into the parapharyngeal space (PPS). In two patients (*n* = 2/18, 11.1%), the retropharyngeal space was also involved. The mean diameters of the lesions were as follows: transverse = 6.12 ± 3.21 cm; axial = 5.34 ± 3.08 cm; and oblique = 4.84 ± 2.87 cm. In line with the current standard of care, all cases were managed surgically. The most commonly employed approach was transcervical (*n* = 6/18, 33.3%). An exclusive transoral resection was performed in two patients (*n* = 2/18, 11.1%). A transparotid approach was still used in two cases (*n* = 2/18, 11.1%), whereas total parotidectomy was performed for another two patients, with mandibular condylotomy required in one case. A combined transoral–transcervical resection was employed for two other patients. In one case, a lip-split approach with mandibulectomy was deemed appropriate for radical resection. Most tumors were histologically benign. Pleomorphic adenoma was the most common diagnosis (*n* = 8/18, 44.4%), followed by lipoma (*n* = 6/18, 33.3%). One case each of angiolipoma and schwannoma (*n* = 1/18, 5.6%) was also reported. Malignant tumors were rare: one liposarcoma (*n* = 1/18, 5.6%) and one follicular dendritic cell sarcoma (*n* = 1/18, 5.6%) were identified. The mean follow-up duration was 19.3 ± 21.2 months. The mean preoperative AHI was 56.42 ± 24.32, which significantly decreased to 15.40 ± 11.93 at the follow-up polysomnography (mean difference: 41.02 ± 7.05, *p* = 0.002). Two patients experienced clinical recurrence of pleomorphic adenoma, both of whom underwent surgical re-resection. In one case, this required facial nerve sacrifice and adjuvant radiotherapy. No recurrence was observed among patients with malignant neoplasms.

## 4. Discussion

The PPS is a virtual, inverted cone-shaped space extending from the skull base superiorly to the hyoid bone inferiorly, and is bounded medially by the buccopharyngeal fascia, posterolaterally by the carotid sheath, and posteromedially by the retropharyngeal space. A fascial plane running posteriorly from the styloid process to the tensor veli palatini muscle divides the PPS into pre-styloid and post-styloid compartments. The complex anatomical composition of the PPS contributes to the histopathological diversity of its tumors. According to the existing literature, salivary gland tumors predominate in the pre-styloid region, while neurogenic tumors are more frequently found in the post-styloid compartment [30]. Due to the absence of rigid anatomical boundaries, PSTs generally become symptomatic after significant growth. The most commonly presented symptom is neck swelling, whereas little is known about the possible relationship between PSTs and the development of OSAS. To date, the literature is still limited to a few reports and case series, preventing researchers from conducting robust systematic reviews or meta-analyses. Nevertheless, this review represents the first comprehensive synthesis of the available evidence. Specifically, we aimed to highlight the importance of a patient-tailored approach to OSA. To date, personalized medicine has been crucial to comprehensively approach OSA and effectively account for the atypical presentations potentially caused by the extrinsic compression of the UA by parapharyngeal space tumors (Table 2). Overall, we observed a marked male predominance (M:F = 17:1). This contrasts with the existing literature, where a slight female predominance is more commonly reported [31]. One possible explanation is that PST patients who develop OSA may possess different anatomical traits that predispose them to UA obstruction (e.g., neck circumference, etc.) which are more frequently observed in males. In contrast, other demographic data, such as the mean age at presentation, were consistent with findings from previously published series [32]. Consistently with the existing literature [11,12], most patients presented with benign lesions (*n* = 16/18, 88.89%), with pleomorphic adenoma being the most frequently reported histology (*n* = 8/18, 44.4%). Interestingly, we observed a relatively high prevalence of benign lipomas originating from the PPS soft tissues (*n* = 6/18, 33.3%). Since lipomas are infrequently found in the PPS [33], we believe that their prevalence among patients with OSA may be higher due to their slow-growing indolent course. As these lesions expand, they may follow paths of least resistance, gradually narrowing the UA and contributing to OSA symptoms. Two cases of malignant PSTs presenting with OSA have been reported to date. Li F. et al. [16] reported the case of a follicular dendritic cell sarcoma originating from the left PPS extending to the homolateral submandibular region. The neoplasm was addressed surgically through a combined transoral–transcervical approach and was still free of disease at follow-up. A liposarcoma of the PPS was first reported by Li H. et al. [18], which was treated with exclusive surgery without the need for adjuvant therapy, and with no signs of recurrence during follow-up. To date, surgery remains the mainstay of treatment, although surgical approaches are highly individualized based on the tumor size, location, and relationship with adjacent structures. Consistently with the existing literature [34], the transcervical approach was the most frequently used in our review (*n* = 6/18, 33.3%). By transecting the stylohyoid muscle, along with the stylomandibular ligament, this approach provides an adequate surgical field for most PSTs. Drawbacks include cosmetic sequelae due to postoperative scarring and potential injury of the marginal mandibular branches of the facial nerve. In contrast, the transoral approach offers a scar-free alternative and reduces the risk of facial paresis, although it is only feasible in patients with sufficient oral exposure. Additionally, transoral access may not allow for adequate control of intraoperative bleeding, particularly from major vessels such as branches of the external carotid artery which may be inadvertently injured during transoral dissection. In such cases, an unplanned conversion to a transcervical approach may be required. Minimally invasive techniques, including endoscopic and robot-assisted approaches, have recently been introduced, showing promising results in selected cases [35]. Nevertheless, no cases of PSTs presenting with OSA managed via transoral robotic surgery (TORS) have been reported to date, enhancing the need for further research in this area. The role of non-surgical treatments is still confined to selected cases. Given the predominantly benign nature of PSTs, radiation therapy (RT) is generally reserved for cases with positive surgical margins or clinical recurrence, as well as symptomatic patients who are unfit for surgery [36]. In our review, only one patient was referred for adjuvant RT and surgical re-excision after experiencing the recurrence of a pleomorphic adenoma originating from the deep lobe of the left parotid gland [24]. A radical parotidectomy with sacrifice of the left facial nerve was necessary, after which the patient achieved recurrence-free survival at follow-up. Finally, in patients with benign tumors and a high surgical risk, a watchful waiting approach may be considered, given the proximity of the PPS to critical structures [11].

Preoperatively, most patients presented with severe OSA, which was consistently and significantly reduced at follow-up (mean difference: 41.02 ± 7.05/h, *p* = 0.002). During sleep, the UA is more susceptible to collapse due to reduced muscle tone, especially in deep and REM sleep. Parapharyngeal space tumors can affect the critical closing pressure (Pcrit) of the UA, contributing to the development of sleep apneas. Notably, other non-anatomical factors, such as genioglossus muscle responsiveness, the arousal threshold, and respiratory control stability, also influence the phenotype of OSA, possibly delaying the presentation of sleep-disordered symptoms [37]. In this context, the literature homogeneously confirmed the effectiveness of surgery in improving the apnea index during sleep. Unfortunately, most of the included studies lacked the reporting of other key polysomnographic parameters (e.g., the apnea/hypopnea ratio, oxygen saturation indices, positional characteristics), thus preventing us from speculating on the potential contribution of non-anatomical factors in these patients.

We herein present the case of a 60-year-old female with a PPS tumor presenting with severe obstructive sleep apnea and daytime somnolence. The patient underwent surgical excision via a transcervical approach (Figure 2), with considerable improvement in the apnea index and sleep quality. Notably, the normalization of polysomnographic parameters was not observed in the first postoperative period but rather at a follow-up polysomnography performed four months after surgery (Table 1). This delayed improvement is a novel observation not previously described in the literature. We hypothesize that postoperative soft tissue edema may transiently increase Pcrit, maintaining airway collapsibility in the immediate postoperative period. Furthermore, inflammation has been previously linked to dysfunctional muscular response, possibly worsening the contribution of non-anatomical factors [38]. This could enhance the possible applications of de-escalated surgical approaches in order to enable faster recovery from sleep apnea in the postoperative period. Notably, Pirovino et al. reported complete resolution of daytime somnolence- and apnea-related symptoms within one month after transoral PPT resection, underscoring a potentially quicker improvement in airway collapsibility with this technique [19]. It is possible that minimizing intraoperative tissue manipulation reduces postoperative soft tissue edema and local inflammation, thereby facilitating a faster recovery compared to traditional open approaches. Interestingly, we also noted a paradoxical increase in the snoring rate at a 4-month follow-up compared to preoperative values. The link between snoring and OSA severity is still debated. Although several studies report a direct association, other findings suggest a more complex relationship [39]. Recently, Hong et al. reported no significant correlation between the AHI and snoring in a large observational cohort and noted a lower snoring rate in patients with very severe OSA compared to those with milder forms [40]. Understanding this paradox requires consideration of the various factors that influence snoring, such as the route of breathing, vibrating sites, sleep position, and apnea–hypopnea ratio. The complete airway obstruction in apnea events leads to the total cessation of airflow, which eventually causes snoring reduction. Following treatment, improved airway patency may cause a shift from apneic to hypopneic events, which can eventually increase the snoring rate. Future directions should entail performing larger observational studies with wider cohorts and complete assessment of polysomnographic data including the apnea–hypopnea ratio in order to strengthen these preliminary assumptions.

## 5. Conclusions

Obstructive sleep apnea syndrome represents a widespread condition with several implications for both patients’ quality of life and the healthcare system as a whole. In the era of personalized medicine, the management of OSAS should be tailored to each patient’s specific characteristics and upper airway anatomy. While intrinsic oropharyngeal collapse is the most common cause of obstruction in OSAS patients, clinicians must also consider rare causes, such as extrinsic compression potentially due to head and neck masses. We present a rare case of severe OSAS caused by extrinsic compression of the PPS, which was successfully treated via a transcervical surgical approach. In such cases, accurate diagnosis is essential. A thorough physical examination, including an assessment of the upper aerodigestive tract, is crucial to personalize treatment, identifying possible extrinsic causes that may not respond to conventional treatments.

## Figures and Tables

**Figure 1 jpm-15-00331-f001:**
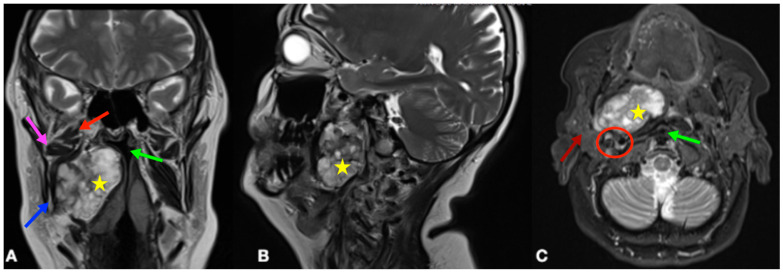
Magnetic resonance imaging of the head and neck. Coronal view (**A**), sagittal view (**B**), and axial view (**C**). Most relevant anatomical landmarks are highlighted. Blue arrow: angle of the mandible; pink arrow: inferior belly of the external pterygoid muscle; red arrow: superior belly of the external pterygoid muscle; green arrow: narrowed pharyngeal lumen; brown arrow: parotid gland; red circle: carotid space; yellow star: parapharyngeal tumor.

**Figure 2 jpm-15-00331-f002:**
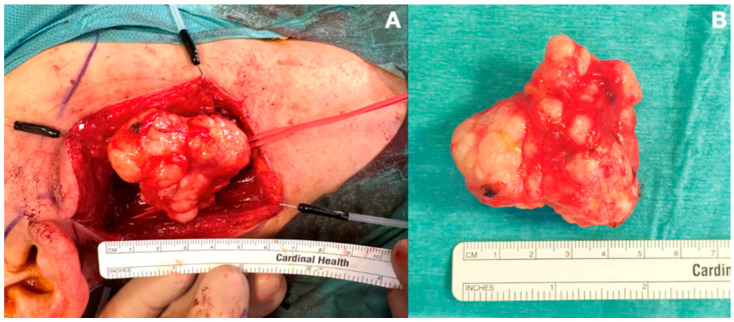
Intraoperative transcervical approach (**A**). Final specimen (**B**).

**Figure 3 jpm-15-00331-f003:**
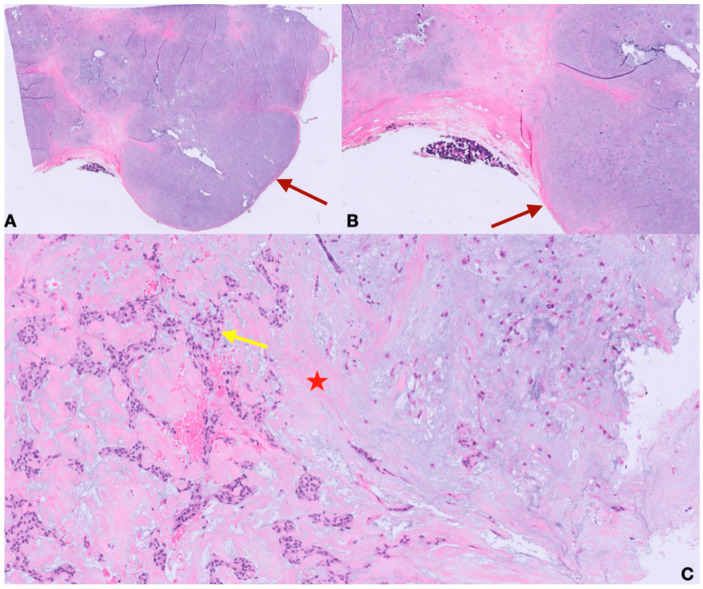
Hematoxylin–eosin-stained specimen at different magnifications (**A**–**C**). A pseudo-encapsulated mass (**A**,**B**, brown arrow) with predominant proliferation of myoepithelial (**C**, yellow arrow) and chondromatous components (**C**, red star) was observed, consistent with a diagnosis of pleomorphic adenoma.

**Table 1 jpm-15-00331-t001:** Trend of polysomnographic parameters. AHI: apnea–hypopnea index; ODI: oxygen desaturation index; CT90: relative time with oxygen saturation < 90%; ESS: Epworth Sleepiness Scale.

Timepoint	AHI (Supine/Not Supine)	ODI	Nadir SpO_2_	CT90	Mean Nadir SpO_2_	Snoring Rate	ESS
Preoperative	77.1/h (77.3/76.8)	73.3/h	68.0%	24.7%	87.4%	29.3%	12/24
2 months	43.5/h (43.5/43.5)	42.5/h	72.0%	34.2%	87.5%	58.2%	8/24
4 months	13.7/h (23.2/8.5)	14.6/h	81.0%	17.9%	87.1%	63.4%	6/24

**Table 2 jpm-15-00331-t002:** Included studies. M: males; F: females; L: left side; R: right side; NR: not reported; NED: no evidence of disease.

Author	Design	Population	Gender (M/F)	Age (Years)	BMI (kg/m^2^)	Origin (Side)	Size (cm)	Surgical Approach	Histopathology Report	Preoperative AHI	Postoperative AHI	Follow-Up, Duration
Marion F et al. 2019 [14]	Case report	1	1/0	69	26	Parapharyngeal space (R)	12 × 10 × 3.3	Transcervical	Lipoma	34/h	2/h	NED, 8 months
Zhu SJ et al. 2014 [15]	Case series	2	2/0	(1) 33 (2) 34	(1) 22 (2) 19	(1) Deep lobe parotid (L) (2) Deep lobe parotid (L)	(1) 3 × 4.5 (2) 4 × 5	Transoral	(1,2) Pleomorphic adenoma	(1) 60/h (2) 24/h	NR	(1) Recurrence, 9 years (2) NED, 2 years
Li F et al. 2024 [16]	Case report	1	1/0	22	NR	Parapharyngeal space (L)	3.2 × 2.1 × 4.5	Transoral	Follicular dendritic cell sarcoma	NR	NR	NED, unspecified
Loudghiri M et al. 2023 [17]	Case report	1	1/0	44	NR	Parapharyngeal space (L)	6 × 9.2 × 4	Combined transoral–transcervical	Lipoma	NR	NR	NED, 1 month
Li H et al. 2013 [18]	Case report	1	1/0	30	NR	Parapharyngeal space (L)	7 × 7 × 6	Transcervical	Liposarcoma	NR	NR	NED, 6 months
Pirovino CA et al. 2018 [19]	Case report	1	1/0	54	30.9	Parapharyngeal space (R)	6 × 4 × 2	Transoral	Lipoma	42/h	11/h	NED, 4 months
Luczak K et al. 2015 [20]	Case report	1	1/0	75	NR	Para- and retropharyngeal space (R)	8.5 × 5.8 × 7.2	Transcervical	Lipoma	NR	NR	NED, 14 months
Casale M et al. 2012 [21]	Case report	1	1/0	70	38	Para- and retropharyngeal space (L)	9 × 6	Transcervical	Lipoma	65/h	31/h	NED, 1 month
Alobid I et al. 2004 [22]	Case report	1	1/0	47	NR	Parapharyngeal space (L)	3.5 × 3 × 8	Cervical–transparotid	Angiolipoma	72/h	<10/h	NED, 54 months
Pellanda A et al. 2003 [23]	Case report	1	1/0	53	25.3	Parapharyngeal space (R)	9 × 4 × 5	NR	Lipoma	32.35/h	8.9/h	NED, 2 months
Giddings CE et al. 2005 [24]	Case series	2	1/1	(1) 35 (2) 67	(1) NR (2) 36	(1) Deep lobe parotid (L) (2) Deep lobe parotid (R)	(1) NR (2) 4 (transverse diameter)	(1) Total parotidectomy (2) Total parotidectomy with mandibular condylotomy	(1,2) Pleomorphic adenoma	(1) NR (2) 93.2	(1) NR (2) 29.5	(1) Recurrence, 2 years (2) NED, 2 years
Adams AJ et al. 2008 [25]	Case report	1	1/0	54	NR	Parapharyngeal space (R)	6 × 5 × 3.5	Lip-split with mandibulotomy	Pleomorphic adenoma with chondroid metaplasia	NR	NR	NED, unspecified
Morariu M et al. 2012 [26]	Case report	1	1/0	42	NR	Deep lobe parotid (R)	7 × 6 × 4	Cervical–transparotid	Pleomorphic adenoma	NR	NR	NED, 6 months
Mulla O et al. 2013 [27]	Case report	1	1/0	60	26	Deep lobe parotid (R)	6.7 × 3 × 6.8	NR	Pleomorphic adenoma	NR	NR	NED, 6 months
Wang AY et al. 2014 [28]	Case report	1	1/0	26	NR	Deep lobe parotid (R)	7 × 5.5 × 3.8	Transcervical	Pleomorphic adenoma	NR	NR	NED, unspecified
Walshe P et al. 2002 [29]	Case report	1	1/0	72	NR	Parapharyngeal space (L)	NR	Transcervical	Shwannoma	10	NR	NR

## Data Availability

The data presented in this study are available on request from the corresponding author due to the patient’s privacy.

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
