# Peer review of "Tumors of the Parapharyngeal Space Presenting with Obstructive Sleep Apnea: A Case Report and Literature Review"

_jpm, 2025, doi:10.3390/jpm15080331_

Round 1
Reviewer 1 Report
Comments and Suggestions for Authors
This is interesting study due to the rarity of this observation (OSAS and PPT). The slow decrease in AHI after surgery with an external approach is interesting. Such a phenomeon could be expected more with intraoral access??? Do other authors confirm this fact? Expain it, please in the text.
Author Response
We sincerely thank the reviewer for their valuable suggestions, which have helped us improve the clarity and overall quality of our manuscript.
Comment:
The slow decrease in AHI after surgery with an external approach is interesting. Such a phenomeon could be expected more with intraoral access??? Do other authors confirm this fact? Expain it, please in the text.
Response:
We thank the reviewer for this insightful suggestion. We fully agree that the delayed improvement in AHI observed in this patient who underwent a transcervical approach for the removal of a parapharygeal space tumor (PPT) represents a key novelty of our manuscript. With the aim of facilitianting quicker postoperative improvement of polysomnographic parameters, de-escalating surgical invasiveness may be justified in selected cases. In support of this, a recent study by Pirovino et al. reported rapid relief from daytime somnolence within one month following transoral resection of a PPT, which reinforces this hypothesis. Unfortunately, robust evidence is still lacking, thus larger observational comparative studies are needed to address this issue.
We have updated the manuscript accordingly in response to your suggestion.
Again thank you for considering our manuscript for publication
Sincerely
The Authors
Reviewer 2 Report
Comments and Suggestions for Authors
A good case write up discussing on the PPS tumour causing OSA. There are several minor revision required as in yellow highlight within the attached text
- Please labels related structures of importance, both for MRI image and Operative photos.
- Please do describe in details the steps of transcervical approach for the case.
Thank you

Author Response
We thank the reviewer for their valuable suggestions, which have helped us improve the clarity and overall quality of our manuscript.
Below, we provide a point-by-point response to the reviewer’s comments:
1) Please labels related structures of importance, both for MRI image and Operative photos.
We appreciate this suggestion and fully agree that proper labeling enhances both reader understanding and the educational value of the manuscript. Accordingly, we have updated Figures 1 and 3 to include labelling of the main anatomical landmarks and relevant pathological features.
2) Please do describe in details the steps of transcervical approach for the case.
We thank the reviewer once again for this helpful comment. A clear description of the surgical steps is indeed crucial, particularly for readers who may be less experienced with this approach. As suggested, we have now included a detailed, step-by-step description of the transcervical procedure in the main text.
Additionally, we have revised the manuscript to address the minor comments and highlighted in the attached file.
Again, we thank you for considering our manuscript
Sincerely
The Authors